# Biological Effects of High-Voltage Electric Field Treatment of Naked Oat Seeds

**Wenqian Xu [1], Zhiqing Song [1,\*], Xinyu Luan [1], Changjiang Ding [1], Zhiyuan Cao [1] and Xiaohong Ma [2]**

[1] College of Science, Inner Mongolia University of Technology, Hohhot 010051, China; wenqianxu0819@163.com (W.X.); luanxy21@sina.com (X.L.); ding9713@163.com (C.D.); czyimut@163.com (Z.C.)

[2] Inner Mongolia Shuangqi Pharmaceutical Co., Ltd., Hohhot 010010, China; xiaohong141@126.com

\* Correspondence: zqsong@imut.edu.cn; Tel.: +86-471-657-5863

**Abstract:** In order to study the mechanism of high-voltage electric field (HVEF) biotechnology, corona discharge produced by a multi-needle-plate HVEF was used to treat naked oat seeds, each treatment dose was divided into two groups, one group was covered with a petri dish cover, the other group was directly exposed to HVEF without a petri dish cover. The scanning electron microscope (SEM) results show that the etching degree of the uncovered group was more serious than that of the covered group, it indicates that ion wind etching has a greater impact on the micro-morphology of seed coat, being covered can effectively reduce the etching degree of discharge plasma on seed. Fourier Transform infrared spectroscopy (FTIR) of the seed coat shows most of the HVEF treatment group can form a new absorption peak at 1740 cm$^{-1}$, which is closely related to the hydrophilicity of the seed. Comprehensive analysis shows that HVEF treatment can improve the hydrophilicity of seeds, whether they are covered or not. Being covered can reduce the degree of etching of the seed coat, but increase the hydrophilicity of the seed, indicating that the non-uniform electric field has a greater impact on the hydrophilicity of the seed. Our study showed that ion wind had an effect on the micro-morphology of seeds, but this effect didn't translate into a macroscopic effect. This study provides ideas and experimental data support for the study of the biotechnological mechanism of HVEF.

**Keywords:** HVEF; naked oat seeds; hydrophilic; seed coat microstructure

## 1. Introduction

The naked oat is a kind of grain-grass-type crop that is cold-loving, drought-resistant and with a strong barren resistance, and it has the highest protein content in cereal grains and is a health-care grain crop [1]. It plays an important role in improving people's diet and ecological construction of agriculture and animal husbandry. With the improvement of people's living standards, the food demand brought about by the rapid increase in population has been increasing, the animal husbandry industry has developed rapidly, the demand for forage has soared and the shortage of forage is more obvious. China's main naked oat producing areas are combined with semi-agricultural and semi-pastoral areas to breed drought-resistant, barren-resistant, high-yield and high-quality varieties. This is more important as a new naked oat variety is being used for both grain and grass. Therefore, it is very important to study the biological effects of high-voltage electric field (HVEF) on naked oat seeds to improve the quality and yield of naked oat.

Multi-needle-plate HVEF is a non-uniform electric field which can produce corona discharge. Multi-needle-plate HVEF is commonly used in the food thawing and drying industry. Researchers have

thawed frozen pork, tuna, shrimp and tofu, and found that HVEF thawing has many advantages [2–5] such as reduced thawing time, food quality preservation, microbial growth inhibition, reduced energy consumption and so on. It was also found that the drying speed and quality of tomato, carrot, sea cucumber, mushroom and potato chips were greatly improved by multi-needle-plate HVEF compared with the control [6–10], which was caused by the synergistic effect of ion wind and the non-uniform electric field. Ion wind had a greater influence on the drying characteristic parameters, and the non-uniform electric field had a greater influence on the quality parameters under the drying process [11]. It has been found that HVEF can promote the formation of callus and obtain highly efficient and high-yielding mutant bacteria [12], and that HVEF biotechnology is a new type of physical mutagenesis technology with the advantages of a simple device, obvious biological effects, environmental friendliness, etc., and obtained Chinese patent authorization [12,13]. The biological effect of the HVEF is generated by the synergistic combination of various factors. The analysis of the physical essential characteristics of the HVEF shows that the two main factors can affect the organism, one is the high-voltage non-uniform electric field, and the other is the discharge plasma. However, the role or contribution rate of these two physical factors has not been clarified.

Studies have shown that discharge plasma can increase the germination, growth and yield of various plant seeds such as pea, wheat, tomato and *Andrographis paniculata* [14–17], it can improve the hydrophilicity of the surface of soybean seeds, thereby increasing the germination speed of seeds [18], it etches the wheat seed coat, which leads to an increase in water absorption capacity [19], it can reduce the apparent contact angle of *Erythrina velutina* (mulungu) seeds and change the hydrophilicity of seeds [20]. The interaction between HVEF and an organism is very complex, in order to explore the role and contribution rate of the two factors in HVEF, when naked oat seeds were treated, one group was exposed directly, the other group was covered with a petri dish cover (blocking the influence of ion wind, the result can be approximated by the electric field single factor effect). The biological effects mechanism of HVEF was discussed by analyzing the changes of hydrophilicity, germination and reactive oxygen species (ROS) content, seed coat microstructure and Fourier Transform infrared spectroscopy (FTIR) of seeds in different treatment groups.

## 2. Materials and Methods

### 2.1. Seed Sample

The naked oat seeds used in the experiment are Ba You No. 1. Only healthy seeds without visible defects were selected for treatment. The naked oat seeds were quickly rinsed three times with distilled water, washing away the surface impurities and dust, filter paper was used to remove the surface moisture, and then the seeds dried naturally. They were stored in a glass culture flask with a vented cover and placed in a refrigerator at 0–5 °C before use.

### 2.2. Experimental Equipment

The HVEF experimental setup used in this study is shown in Figure 1. The power supply was alternating current (AC) with a discharge voltage range of 0–50 kV and frequency of 50 Hz. The electrode system was a multi-needle-plate array with needle spacing of 4 cm and needle length of 2 cm. The length and width of the electrode system were 60 cm and 40 cm, respectively, eliminating the boundary effect. The grounding end was a plane aluminum plate with length and width of 100 cm and 45 cm, respectively, and the distance from the tip of the electrode to the grounding end was 4 cm. The selected seeds were evenly placed in a petri dish with a diameter of 10 cm, and 100 seeds per dish. The samples were then divided into two groups, one group was treated with no petri dish cover and the other group was covered with a 1 mm polypropylene petri dish cover. The above two groups of samples were treated with voltages of 0 kV (control group, marked as CK), 4 kV, 8 kV, 12 kV, 16 kV and 20 kV, the treatment time was 10 min and each treatment was repeated three times. The direct exposure group was recorded as 0 kV (CK), 4 kV, 8 kV, 12 kV, 16 kV and 20 kV, and the covered petri

dish group was recorded as 0 kV + covered (CK), 4 kV + covered, 8 kV + covered, 12 kV + covered, 16 kV + covered, 20 kV + covered. The actual measured value of 0 kV and 0 kV + covered is the same. Temperature and relative humidity were 24 ± 2 °C and 30% ± 5%, respectively, during the process.

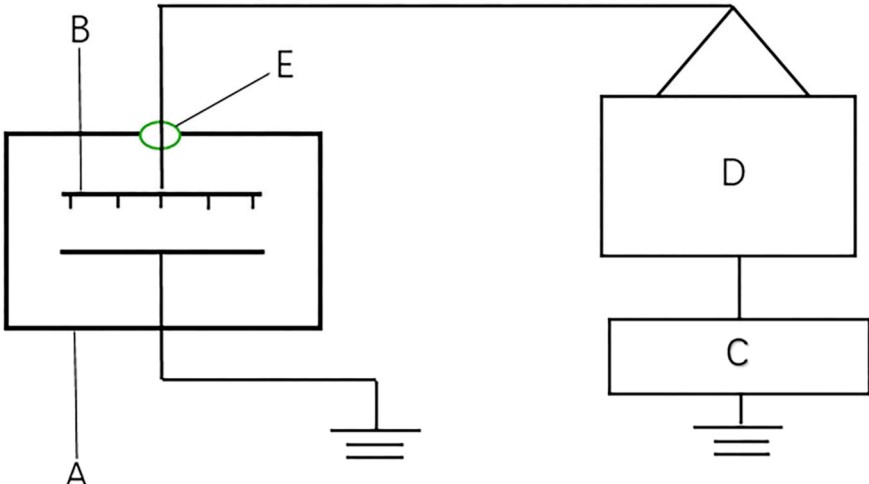

**Figure 1.** High-voltage electric field (HVEF) device. Note: A is the ultra-clean workbench; B is the electrode system; C is the control unit; D is the high-voltage power supply; E is the high voltage insulation tube.

### 2.3. Floating Rate and Water Absorption Test

The treated seeds were accurately weighed and marked with one ten thousandth of an electronic balance, and 40 mL of deionized water was added into the petri dish to count the number of seeds floating in each group. After soaking for 20 h, the surface moisture of the seed was dried with filter paper and the seed was weighed after water absorption. The floating rate (FR%) and water absorption rate (WAR%) are calculated by the following equations:

$$\text{FR\%} = \frac{N_f}{N_s} \times 100\%, \tag{1}$$

$$\text{WAR\%} = \frac{M_1 - M_0}{M_0} \times 100\%, \tag{2}$$

where $N_f$ is the number of seeds floating in the water, $N_s$ is the number of total soaked seeds, $M_0$ is the weight of the seed before water absorption, $M_1$ is the weight of the seed after water absorption.

### 2.4. Seed Height and Fresh Weight Test

The seeds were placed in a petri dish containing three layers of filter paper, and distilled water was added to each dish every other day to keep the water sufficient. The seeds were placed in a light incubator at a constant temperature of 26 °C. After the seeds were germinated, the light incubator was irradiated with 140 Lux light, continuous illumination for 14 h and darkness for 10 h. In each petri dish, 15 seedlings were randomly selected to measure seedling height and fresh weight after 10 days.

### 2.5. Ion Wind Speed Test

The ion wind speed under different experimental conditions was measured using a thermal anemometer probe (405i, Ruice Electronics Technology Co., Ltd., Guangzhou, China). Each experiment was repeated three times independently and averaged.

### 2.6. Infrared Spectroscopy Test

The dried seed coat was ground into a powder using an agate mortar, and the sample powder and the potassium bromide were uniformly mixed at a ratio of 1:500 and filtered with 80 target quasi-sieves. The powder was pressed into the disk using a tablet machine (HY-12, Tianguang Spectrometer Co., Ltd., Tianjin, China). Samples were scanned using a Fourier transform infrared spectrometer (Nicolet iS10, Thermo Nicolet Corporation, Madison, WI, USA).

### 2.7. Scanning Electron Microscopy Test

The sample to be observed was sprayed with a metal film and then adhered to the sample stage with conductive double-sided tape. The seed coat of the electric field treatment surface was observed and photographed by scanning electron microscope (S3400, Hitachi Corporation, Tokyo, Japan).

### 2.8. Seedling ROS Content Test

From the same treated seedlings, 3 g was randomly selected on the third and seventh day of inoculation, and the ROS was measured after the ice bath was ground. The substrate used for the assay was 2′,7′-Dichlorodihydrofluorescein diacetate. The sample to be tested was reacted with 10 mM CM-H2DCFDA at 25 °C for 30 min in the dark according to the method of Babu et al. [21]. The sample was then placed in a 96-well plate containing 100 μL of distilled water in the dark, then put into a microplate reader (SPECTRAMAX I3, Molecular Devices(MD) Corporation, USA). The excitation intensity of 530 nm was detected by excitation with an excitation light of 485 nm. This was repeated six times or more.

### 2.9. Statistical Analysis

All treatments were repeated at least three times. Data from this study were recorded as the mean value ± standard deviation. The SPSS statistical software (Version 16.0) and one-way analysis of variance (ANOVA) were used to analyze the difference of data, and $p < 0.05$ was considered as a significant difference.

## 3. Results

### 3.1. Effect of HVEF on Water Absorption and Hydrophilicity of Naked Oat Seeds

The water absorption of naked oat seeds after treatment with HVEF is shown in Figure 2. It can be seen that the water absorption rate of seeds with or without covered treatment increased when the voltage was 4 kV, 12 kV, 16 kV and 20 kV. However, when the voltage was 8 kV, the water absorption rate decreased. As a whole, except for the 8 kV uncovered treatment group, the water absorption of the other treatment groups was improved by different degrees compared to the control group, and the water absorption rate of the covered group increased under each voltage. The hydrophilicity of the seed during soaking was as shown in Figure 3. It can be seen that the hydrophilicity of the seed increased with or without covered treatment compared with the control group. As can be seen from Figure 4, the floating rate of seeds after covered treatment was significantly lower than that of the control group. Moreover, compared with the uncovered treatment, the floating rate of seeds after covered treatment was reduced, indicating that the covered treatment can better improve the hydrophilicity of seeds.

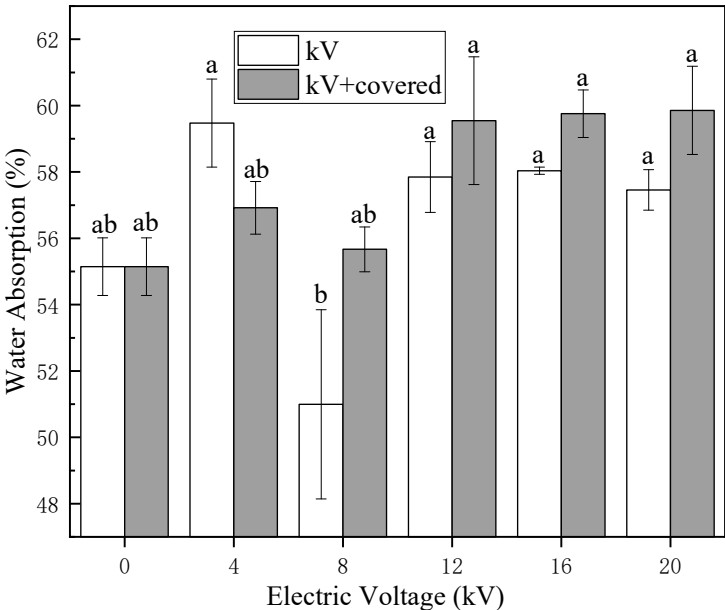

**Figure 2.** Changes in water absorption of naked oat seeds after HVEF treatment (Different letters indicate statistically significant differences between treatments ($p < 0.05$)).

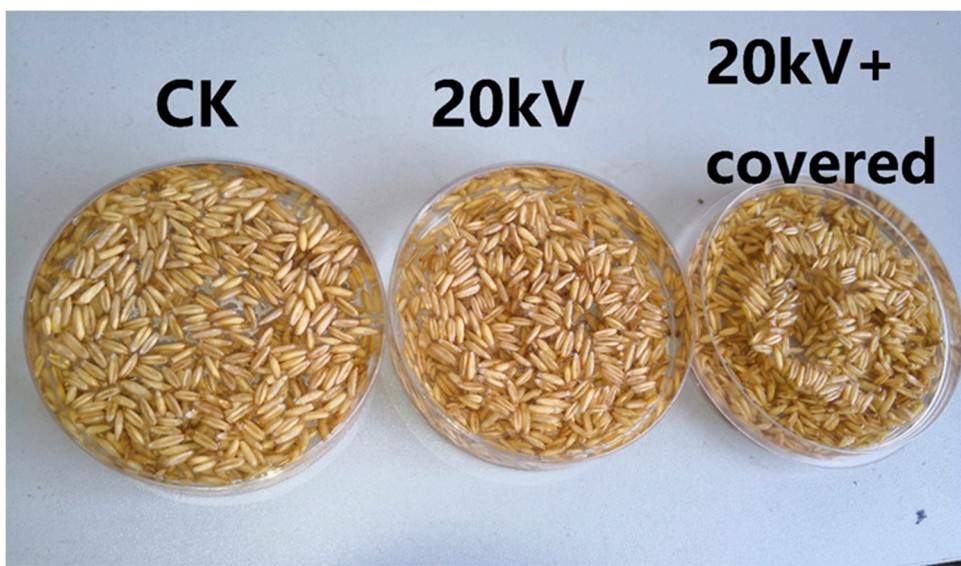

**Figure 3.** Hydrophilic changes of naked oat seeds after HVEF treatment.

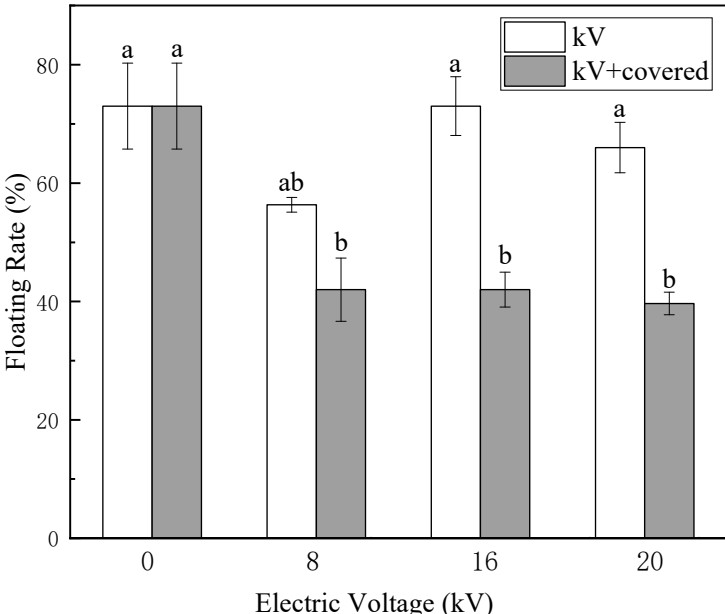

**Figure 4.** Changes in the floating rate of naked oat seeds after HVEF treatment (Different letters indicate statistically significant differences between treatments ($p < 0.05$)).

### 3.2. Effect of HVEF on the Growth of Naked Oat Seedlings

The seedling height and fresh weight of naked oat seedlings after HVEF treatment are shown in Figures 5 and 6, respectively. After HVEF treatment, the seedling height of naked oat seedlings increased by different degrees with or without a cover, and the fresh weight of seedlings also showed an upward trend with the increase of voltage. However, the effect of ion wind on seedling growth is not significant.

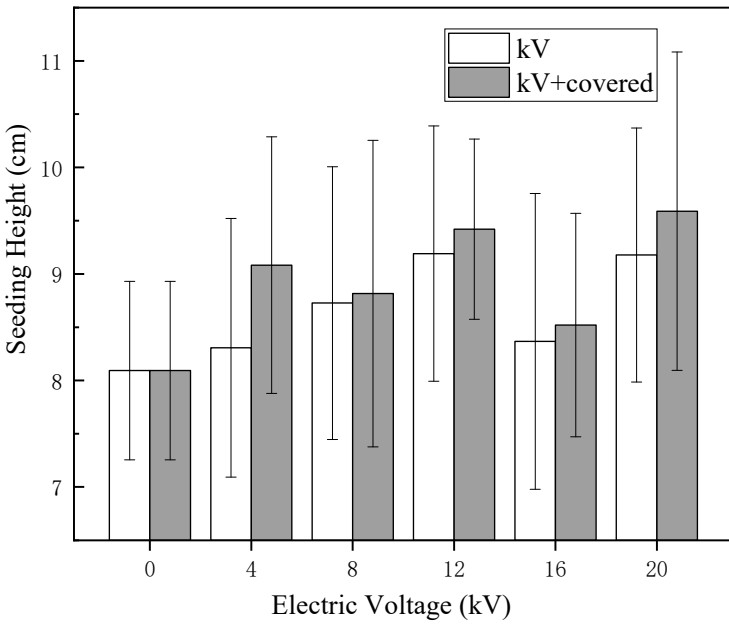

**Figure 5.** Changes in seedling height of naked oat seedlings after HVEF treatment.

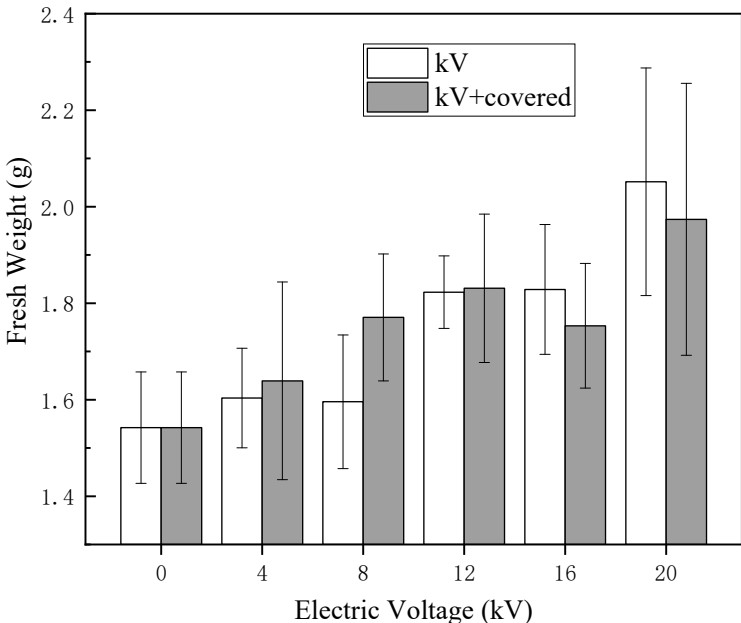

**Figure 6.** Changes in fresh weight of naked oat seedlings after HVEF treatment.

### 3.3. Effect of HVEF on Naked Oat Seed Coat

Figure 7 shows the ion wind speed treated by HVEF under different voltages. It can be seen from the figure that there is a linear relationship between the ion wind speed and the voltage of the uncovered treatment group. However, the ion wind speed of the covered group was nearly zero, indicating that the petri dish cover could effectively block the ion wind.

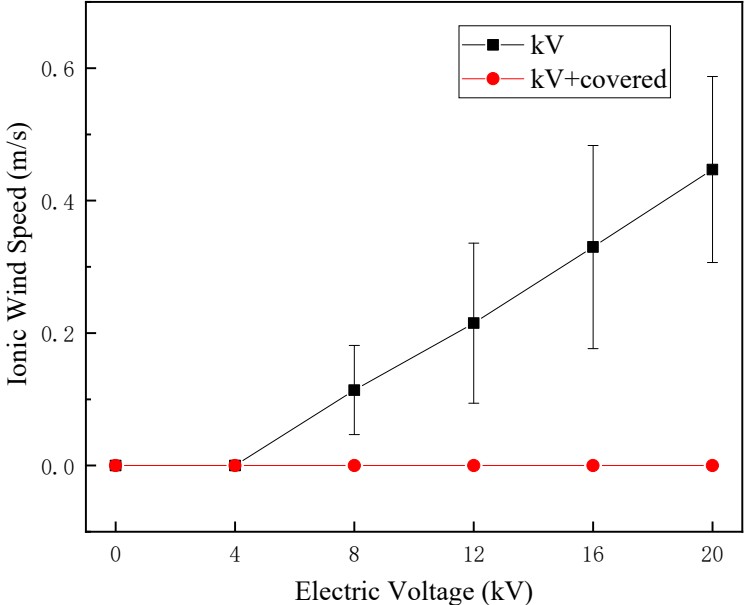

**Figure 7.** Changes of ion wind speed under HVEF.

The microscopic morphology of naked oat seed coat before and after HVEF treatment is shown in Figure 8. The grid structure can be clearly observed in the control group, and the boundaries of these square network structures can be clearly seen. As shown in Figure 8b, after the 8 kV uncovered treatment, small cracks formed on the surface of the seed coat, and it was obvious that surface cracks were more than those in Figure 8c after the covered treatment. It can be seen from Figure 8d,e that the boundary of the surface network structure after the 16 kV and 20 kV uncovered treatment becomes

blurred and difficult to identify, and surface cracks become more obvious with the increase of voltage. It can be seen from Figure 8f that after the 20 kV covered treatment, the surface cracks of the seed coat were significantly reduced compared with the uncovered group.

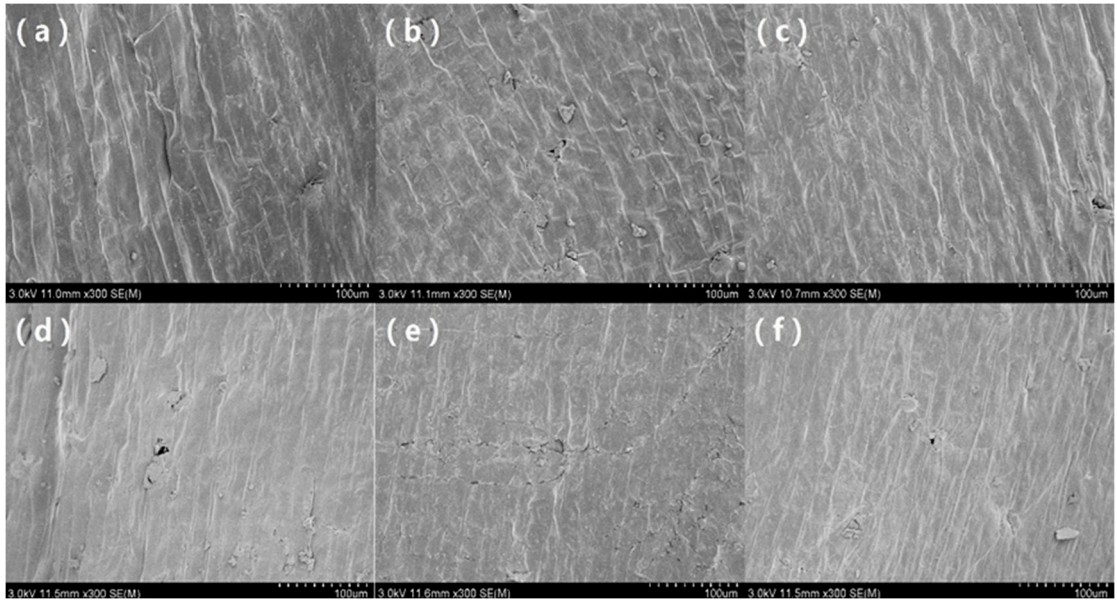

**Figure 8.** Scanning electron microscope (SEM) photos of naked oat seeds under HVEF treatment with x300 magnification. (**a**) CK; (**b**) 8 kV; (**c**) 8 kV + covered; (**d**) 16 kV; (**e**) 20 kV; (**f**) 20 kV + covered.

The FTIR spectra of naked oat seed coat after HVEF treatment is shown in Figure 9a. After the treatment of HVEF, the absorption peak intensity was enhanced at 3281 cm$^{-1}$, 1646 cm$^{-1}$, 1542 cm$^{-1}$, 1412 cm$^{-1}$ and 1035 cm$^{-1}$, regardless of whether or not the petri dish was covered, and the uncovered group was enhanced more than the covered group. For the wave number at 1709 cm$^{-1}$ and 1243 cm$^{-1}$, the absorption peak intensity was weakened. After the uncovered treatment of 16 kV, the peak strength changed more significantly than that of the 16 kV + covered group, indicating that the cover could effectively reduce the surface etching of seeds. Particularly interesting was that, as shown in Figure 9b, except for the 8 kV covered group, the HVEF treatment group formed a new absorption peak near 1740 cm$^{-1}$, and the shape of the absorption peak changed near 1035 cm$^{-1}$. After the treatment of HVEF, some peaks of FTIR were strengthened and weakened by different degrees and formed new absorption peaks, indicating that the chemical etching of naked oat seed coat by HVEF changed the chemical structure of naked oat seed coat.

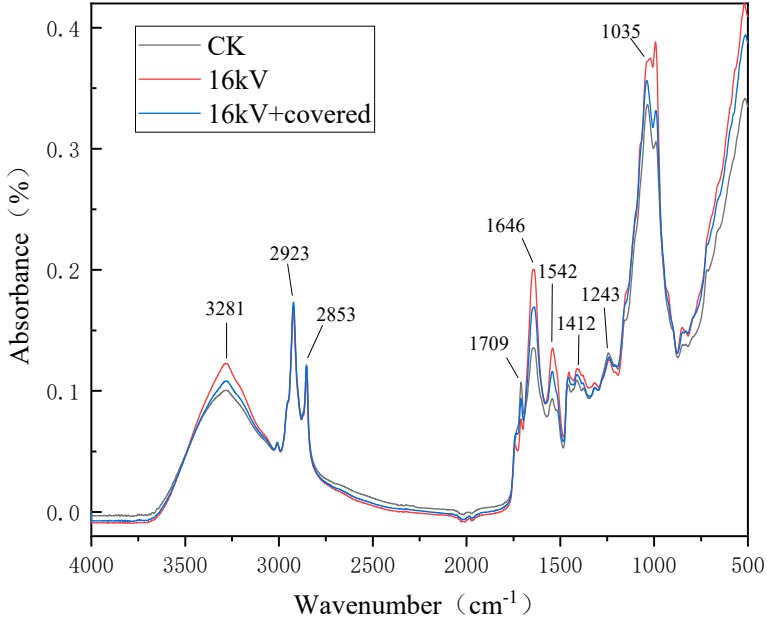

(**a**)

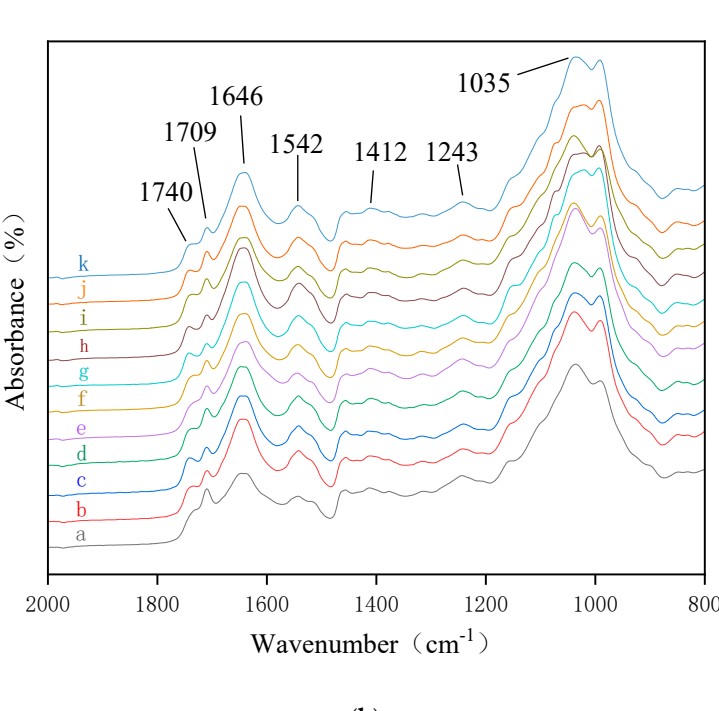

(**b**)

**Figure 9.** Infrared spectra of naked oat seed coat after HVEF treatment: (**a**) Infrared spectra of naked oat seed coat before and after treatment with 16 kV HVEF; (**b**) Infrared spectra of naked oat seed coat before and after treatment with different voltages, wherein a, b, c, d, e, f, g, h, i, j and k represent CK, 4 kV, 4 kV + covered, 8 kV, 8 kV + covered, 12 kV, 12 kV + covered, 16 kV, 16 kV + covered, 20 kV, 20 kV + covered, respectively.

*3.4. ROS Content Changes Before and After Treatment with HVEF*

As shown in Figure 10, the ROS content after treatment on the third day after inoculation was lower than that of the control group, and the ROS content was slightly higher than that of the covered treatment group after uncovered treatment. On the seventh day after inoculation, the ROS content after

treatment was lower than on the third day, the ROS content of the treatment group was significantly lower than that of the control group and the ROS content of the uncovered treatment group was slightly higher than that of the covered group.

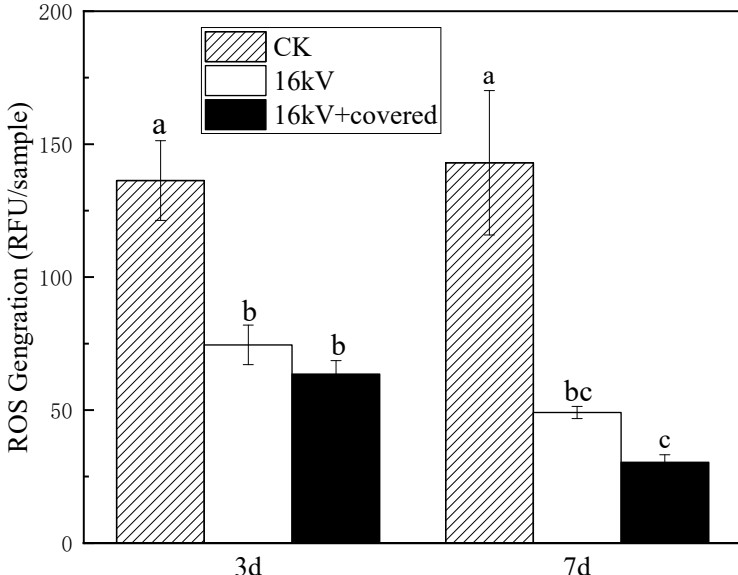

**Figure 10.** Reactive oxygen species (ROS) content changes of seedlings after HVEF treatment (Different letters indicate statistically significant differences between treatments ($p < 0.05$)).

## 4. Discussion

HVEF biotechnology is a new type of high-efficiency bio-mutation technology, and the biological effect of HVEF is produced by the synergistic combination of various factors [12]. According to the analysis of its physical properties, the biological effect of the HVEF is mainly formed by two factors: one is the action of the non-uniform electric field, and the other is the role of discharge plasma. In HVEF, with the increase of discharge voltage, discharge plasma is produced, accompanied by the dissociation, excitation and ionization of gas molecules. Gas molecules undergo complex physical and chemical reactions, resulting in a large number of reactive oxygen and nitrogen species (RONS). RONS are important signal molecules in life activities and can react with bioactive molecules to change their structure. In HVEF, reactive agents (RAs) consist of active material RONS and a non-uniform electric field, which are the main components of ionic wind. RAs can directly penetrate the treatment body and even produce secondary effects [22]. Wang et al. also observed that RAs chemically etched the surface of seeds by oxidizing the organic components on the seed coat [23]. In order to explore the respective roles and contributions of two factors in the HVEF, when treating naked oat seeds, one group was directly exposed to the HVEF, and the other group was covered with a petri dish cover to block the physical and chemical etching of seeds by ion wind.

The HVEF can change the chemical structure and surface microstructure of naked oat seed coat, whether they are covered or not. This indicates that both the non-uniform electric field and discharge plasma can affect the microstructure of seed coat. It can be seen from the scanning electron microscope (SEM) results that after HVEF treatment with different voltages, the surface of the seeds was etched to have different degrees of cracks, showing stronger hydrophilicity and being more conducive to effective absorption of water and nutrients [23]. According to Figure 7, it can be seen that the ion wind speed of the HVEF without a cover increases with the increase of voltage. According to the microscopic morphology of the seed surface in Figure 8, the HVEF treatment caused the cracking of the naked oat seed coat, and with the increase of the ion wind speed, the surface of the seed coat was strengthened and the water absorption rate of the seed increased. The change in the seed coat after treatment with HVEF is related to the etching action of ion wind [12], and the cover treatment can effectively

reduce ion wind etching. When there is no cover, multi-factor synergy has a greater influence on the microstructure of the seed coat, indicating that the ion wind etch has a greater influence on the micro-morphology of the seed coat than the non-uniform electric field, but this effect is not converted into a macroscopic effect.

Previous FTIR studies show that the wave number at 3281 $cm^{-1}$ was the stretching vibrations of O–H and N–H of polysaccharides, sugar alcohols, glycosides, amino acids and proteins; the wave number at 2923 $cm^{-1}$ was the asymmetric stretching vibrations of the -$CH_3$ of lipids and proteins; the wave number at 2853 $cm^{-1}$ was the symmetrical telescopic vibration of -$CH_2$ from membrane lipid; the wave number at 1709 $cm^{-1}$ was the stretching vibrations of C=O, indicative of cutin and wax; the wave numbers at 1646 $cm^{-1}$ and 1542 $cm^{-1}$ were indicative of protein amide I band and II band, respectively, the amide I band was formed by the stretching vibration coupling of C=O and the C–N group, the amide II band was formed by the stretching vibration coupling of N–H and C–N groups; the wave number near 1412 $cm^{-1}$ was the bending vibration of -$CH_2$, mainly from proteins and nucleic acid; the wave number near 1243 $cm^{-1}$ was the asymmetric stretching vibration of the phosphodiester group, which mainly from the phosphodiester skeleton vibration of nucleic acid and phospholipid in the biofilm; the wave number near 1035 $cm^{-1}$ was the symmetrical telescopic vibration mainly from the C-O-C vibration of phospholipids and sugars in the biofilm [24,25].

The surface of naked oat seeds is hydrophobic, the main components of the seed coat surface are cellulose and wax. Cellulose is hydrophilic and insoluble in water, while wax is hydrophobic. As shown in Figures 3 and 5, a large number of untreated naked oat seeds float on the surface of the water. After treatment by electric field, the number of floating seeds was greatly reduced, the hydrophilicity of naked oat seeds was increased and the hydrophilicity of the covered group increased more obviously, indicating that the non-uniform electric field impacts the hydrophilic seeds more than the ion wind. As shown in Figure 9, the wave numbers at 3281 $cm^{-1}$, 1646 $cm^{-1}$, 1542 $cm^{-1}$, 1412 $cm^{-1}$, and 1035 $cm^{-1}$ were enhanced, indicating that the content of hydrophilic substances such as polysaccharides, sugar alcohols, cellulose and hemicellulose increased, these substances are all hydrophilic, especially hemicellulose and can cause the cell wall to swell, impart fiber elasticity and thus improve hydrophilicity. After the treatment of HVEF, the wave number at 1709 $cm^{-1}$ was weakened, which is similar to the results of Himmelsbach and Wang et al. [23,26], indicating that the wax and oil content decreased, that is, the content of hydrophobic substances decreased, the change of these absorption peaks result in changes in the hydrophilicity of naked oat seeds. The new absorption peak formed near 1740 $cm^{-1}$ of hemicellulose [23], which is conducive to the improvement of seed water absorption. In the results of the water absorption measurement in Figure 2, only the water absorption rate of the 8 kV uncovered group decreased slightly compared with the control, while the absorption peak of this group at 1740 $cm^{-1}$ was not obvious. The water absorption rate of the 8 kV occlusion group did not change significantly compared with the control, this group did not form an absorption peak at 1740 $cm^{-1}$, while the other treatment groups formed a new absorption peak at 1740 $cm^{-1}$. The water absorption rate increased by different degrees compared with the control group, so whether an absorption peak is formed at 1740 $cm^{-1}$ is closely related to the hydrophilicity of the seed, and whether it can be used as a judgment mark for the water absorption rate and hydrophilicity of the naked oat seed needs further research. Moreover, after the treatment of HVEF, the peak value of the FTIR of the uncovered group was more severe than that of the covered group, indicating that the ion wind etches the seed coat more seriously when it is not covered. This verifies that the ion wind etching has a greater influence on the microstructure of the seed coat than the non-uniform electric field.

The seed coat state plays an important role in regulating seed germination and seedling vigor. Studies have shown that etching can improve the hydrophilicity and water permeability of the seed epidermis, promoting its absorption of water and transfer to the endosperm and embryo [23]. According to the results of Meng et al. [27], we found that the moderate destruction of the seed coat by electric field treatment is conducive to seed germination. As shown in Figures 5 and 6, the height and

fresh weight of naked oat seedlings were significantly increased after the treatment of HVEF compared with the control group, which indicates that the HVEF treatment can promote the seedling height of the germinated seedlings regardless of whether or not the germination rate is inhibited. Cramariuc et al. [28] treated potato seeds with an HVEF and found that electric field treatment had a positive effect on potato seed growth and yield (especially seed yield). This may be because the appropriate intensity of electric field treatment increases the permeability of biofilm, enhances the enzyme activity and thus speeds up the growth of seeds and increases the fresh weight. Wang et al. showed that electric field treatment can improve enzyme activity, increase cell membrane permeability, accelerate new cell formation, promote root and bud differentiation and growth and improve seed vigor [29], thereby promoting seedling growth after electric field treatment.

Studies have shown that ROS plays an important role in cell proliferation, differentiation and apoptosis and has the function of signaling molecules. ROS is an important marker connecting physical and biological interpretation of the connection of the electromagnetic causal chain. After HVEF treatment, the ROS generated in the cell matrix enters the cell and interacts with the macromolecules in the cell, thus changing the metabolic activity and genetic characteristics of cells [12]. Numerous studies have shown that ROS is harmful to organisms at high concentrations, and at low concentrations, ROS can act as a second messenger to mediate multiple responses of plants to hormones or environmental stress [30,31]. It can be seen from Figure 10 that after treatment with HVEF, the ROS content of naked oat has been significantly reduced, which is consistent with the research results of Sheteiwy et al. [32]. ROS reduction can be used as seed response to physicochemical etching of HVEF.

## 5. Conclusions

HVEF treatment can increase the content of hydrophilic substances, cellulose degradation and decrease the content of hydrophobic substances such as wax in the seed coat, thus increase the hydrophilicity of the seed. HVEF treatment can form a new absorption peak at 1740 cm$^{-1}$, which is closely related to the hydrophilicity of the seed. Ion wind etching has a greater impact on the micro-morphology of the seed coat, but this effect is not converted into a macroscopic effect, being covered can effectively reduce the etching degree of discharge plasma on seeds, reduce the field intensity received by seeds and make the radiation field received by seeds more uniform. The non-uniform electric field has a greater impact on the hydrophilicity of the seed.

**Author Contributions:** All authors have made significant contributions to this research. W.X. and Z.S. performed the majority of the experiments and wrote the manuscript; W.X., Z.S., X.L., C.D., Z.C., and X.M. contributed to the analysis of the data; and Z.S. designed and supervised the study and checked the final manuscript.

**Funding:** This research was funded by National Natural Science Foundations of China (No. 51767020), Scientific Research Project of Inner Mongolia University of Technology (No.X201417).

**Acknowledgments:** The authors would like to express their gratitude to the anonymous referees for their valuable comments and suggestions. Thanks to Kaidi Xu for his help with the experiment.

**Conflicts of Interest:** The authors declare no conflict of interest.

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
