# Peer review of "Biological Effects of High-Voltage Electric Field Treatment of Naked Oat Seeds"

_applsci, doi:10.3390/app9183829_

Round 1

Reviewer 1 Report

Comments and suggestions are highlighted directly in the manuscript.

Reviewer 2 Report

The authors presented a very interesting research work on the biological effects of electric field and corona discharges exposure to seeds. The work has been planned and described in all the steps, however some physical concepts are presented not clearly and some amends would improve significantly their work. In addition some clarifications on the discussion could improve significantly the paper.

In particular, in the paper the authors refers to the corona as a discharge, correctly in line 39 and  in many parts of the paper they associated the term corona with electric field , however corona is a consequence of electric high field and the two concepts should not be mixed and defined as a field. Could the author explain the use of the adopted terminology?

Did the authors record the relative humidity and temperature values during the experiments?

Abstract

Line 12 high-voltage corona and electric field

Line 15  SEM introduced without introducing the meaning before

Text

Line 39 please remove “partial”. The term "partial discharge" is used on localized part indicating ionization due to defects on the insulating  material or electrical stress.

Line 56 the acronym ROS is used but reactive oxygen and nitrogen species (RONS) is resented only at line 212. I suggest to introduce here for the first use the extended explanation.

Line 61 Seed sample. The authors should clarify the procedure adopted to clean the seeds and the storage technique adopted during refrigeration, e.g. open or close container?

Lines 64-72 The seeds are placed on a 10cm diameter petri dish. However the dimensions of the electrodes are not complete. Are the electrode plates wider than the petri dish in order to avoid a very uneven electric field and corona discharge exposures? These dimensions can affect significantly the electric field lines and consequently have a strongly non uniform distribution of the electric field on the seeds

Line 74 A control group 0kV+covered should have been formally defined and tested. (later it has been used in figure 2)

Please improve clarity of figure 1 and of its caption. In the figure 1, please add symbol of label in D and C. please change arrow orientation. A reader will assume A as the top electrode and not the overall workbench. What is the part E, the high voltage insulation card?

Equations 1 and 2 should be better formatted introducing line spaces before and between them

Line 128 How the authors can explain the significant different trend of water absorption of the 8kV uncovered sample?  If this variance is due to experimental error, how could be the other results confirmed within the interval of confidence? Figures 2 and 3 show a consistent different trend for the 8kV samples.

Figure 3 CK group is labelled in figures but not defined in the text.

Line 145 - 149 and Figure 5. Taking into consideration the fact that both samples (uncovered and covered) showed an effect of the treatment, how can be confirmed that the ionic wind is a contribution factor? Later in lines 229-234 it is stated that ion wind etch has  a greater effect. However not on the height plot.

Line 229 -234 Since no ion wind is present at the seed area for covered samples, how does the author support the effect of it on the fresh weight?

I agree that the ionic wind has an impact of the micro-morphology as shown in figure 8. but this effect is the not translated in macroscopic effect as it cannot be seen in figure 5, 6 and  10.

I suggest the author to reformulate some part of their statements avoiding this contradiction.

Round 2

Reviewer 2 Report

Dears authors

the amends introduced into the paper improved significantly the clarity of your research. well done.